# The effects of the COVID-19 pandemic on Italian primary school children's learning: A systematic review through a psycho-social lens

**Eugenio Trotta**[1], **Gianluigi Serio**[1], **Lucia Monacis**[1], **Leonardo Carlucci**[1], **Chiara Valeria Marinelli**[1], **Annamaria Petito**[2], **Giovanna Celia**[1], **Aurora Bonvino**[1], **Antonella Calvio**[1], **Roberta Stallone**[2], **Ciro Esposito**[1], **Stefania Fantinelli**[1], **Francesco Sulla**[1], **Raffaele Di Fuccio**[3], **Gianpaolo Salvatore**[1], **Tiziana Quarto**[1☯]*, **Paola Palladino**[1☯]*

1 Department of Humanities (DISTUM), University of Foggia, Foggia, Italy, 2 Department of Clinical and Experimental Medicine, University of Foggia, Foggia, Italy, 3 Faculty of Human Sciences, Education and Sport, Pegaso University, Naples, Italy

☯ These authors contributed equally to this work.
* tiziana.quarto@unifg.it (TQ); paola.palladino@unifg.it (PP)

**Data Availability Statement:** All relevant data are within the paper and its Supporting Information files.

## Abstract

The COVID-19 pandemic drastically affected many areas and contexts of today's society, including school and family. Several studies focused on the worldwide effects of school closures on students' learning outcomes, context, and well-being. However, the data emerging from these studies are often inconsistent and fragmentary, highlighting the need of a comprehensive analysis of the phenomenon. This need is especially urgent for the countries with the most severe school closure, like Italy. This systematic review aims to collect the opinions of parents, teachers, and students on: other dimensions of Italian primary school students affected by school closures, beyond academic performance; hypothetical agreement between the opinions of parents, teachers, and students regarding the different effects of school closures on Italian primary school students; possible differences between the effects of school closures on Italian primary school students and the students in other countries. Our search was conducted using PRISMA 2020 guidelines on Web of Science, Pubmed, Scopus, and EBSCOHost. The results obtained from 34 articles revealed a strong concern on the part of all stakeholders involved in learning during the pandemic, with evident negative effects for Italian school students. The constraint on distance learning led to a drastic change in everyone's routine, and a negative emotional change on the part of young students. Parents and teachers generally considered distance learning to be ineffective for the education of their children and students; they encountered technical-practical difficulties in the use of electronic devices for participation in school activities; overall learning deficits on the part of students, especially in mathematics, as confirmed by INVALSI results were also found. The investigation reveals a condition of shared emotional and academic performance difficulty, and a further challenging circumstance for students previously at risk of

**Funding:** The author(s) received no specific funding for this work.

**Competing interests:** The authors have declared that no competing interests exist.

marginalization. Further research in this field is paramount to identify new and adequate recovery strategies.

## Introduction

### Background

9[th] March 2020 represents an unforgettable date in recent Italian history. On that date, the former Italian Prime Minister declared the start of the national lockdown [1]. The pandemic has dramatically changed people's life worldwide, impacting all sectors, with repercussions also for the school sector. Indeed, different countries closed educational institutions to prevent the spread of COVID-19 [2], causing an interruption of regular learning. The same has been done in Italy, where schools remained closed for almost 40 weeks [3, 4]. After some days/weeks of total school interruption, most countries, including Italy, started to adopt distance learning [5, 6]. However, distance learning is a complex process that needs a careful design to lead to effective learning [7]. Due to the unpreparedness that led to the implementation of the distance learning, several studies revealed some achievement gap as an effect of school closure on the students' learning quality at different educational levels [8, 9].

König & Frey [10] conducted a meta-analysis including 18 studies from different countries on primary and secondary school students, showing that school learning tended to be negatively affected by school closures, with a particular impact on primary school students. These results can also be found in Betthäuser et al. [11] and Di Pietro [12], comprising respectively 42 and 39 studies from different geographical areas on primary and secondary education students–even tertiary in Di Pietro [12]. Both meta-analyses agreed that students were unable to recover from the learning loss caused by school closures due to COVID-19. Several studies agreed that school closures mainly affected the most marginalized students, both scholastically [13, 14] and emotionally [15, 16]. These students are identified as vulnerable due to disadvantaged educational, environmental, and social factors, including socio-economic status (SES), cultural differences, learning disabilities and special education needs (SEN). Also, the students included in these categories are accompanied by additional scholastic support compared to their classmates, which is challenging to dispense online, and may thus represent an additional source of difficulties in times of pandemic. This shows how the learning environment is essential to the student's well-being [17–19]. As a consequence of the online transition, students experienced a period of forced isolation at home, away from other places of interest and their peers, with negative psychological consequences [20]. Several studies highlighted a fluctuation in the emotional behavior of students during the pandemic, with a significant increase in anxiety, depression, sadness [21, 22], suicide attempts [23], and behavioral problems such as restlessness and inattention [24].

The debate on the quality of distance learning was also accentuated by the presence of problems related to teaching and assessment. Among the difficulties related to teaching, there are technological complications, such as the presence of adequate support tools, computers, good quality broadband. Another challenge is creating an ideal study environment [25–27]. Furthermore, ensuring the proficiency adequate of both teachers and students, along with their parents, in the use of e-learning instruments and practices is essential [28, 29]. Engzell et al. [30] reported that the Netherlands–whose schools were closed for eight weeks–did not experience such technological problems. However, the study refers to a country with an excellent school performance according to the Organization for Economic Cooperation and

Development (OECD) average, high academic autonomy, and the greatest presence of broadband among European countries. Despite the lack of such technological problems, learning losses were also found in Dutch students, highlighting that low socio-economic background and parental education/support negatively influenced students' learning [31].

Based on the information previously mentioned, it appears that the closure of schools due to COVID-19 and the subsequent distance learning affected students across educational levels worldwide, with implications extending beyond academic performance. It is plausible that similar circumstances have also transpired in Italy. Indeed, according to the Italian school calendar [32], Italian students lost over a year of regular learning during the pandemic. The national institute for the evaluation of the education and training system (INVALSI) annually tests the learning outcomes in language, mathematics, and English of Italian 2nd and 5thgraders. The 2022 INVALSI data, compared with 2019 and 2021 cohorts, revealed a containment of the pandemic effects on learning of Italian language and a general slight worsening in mathematics [33]. Moreover, these data show that students with a linguistic disadvantage achieved lower results compared to an average student of the same age, highlighting the post-pandemic difficulties of students with low SES and a migrant background. However, although these data also consider socio-economic, cultural, and migratory aspects, they do not consider further aspects that may have influenced students' learning outcomes during the pandemic. Several international studies highlighted how the transition to distance learning led to serious emotional consequences and behavioral changes on students [15, 16, 34, 35]. Assessment of distance learning is still a widely debated topic, as digital skills among teachers and parents are heterogeneous–both in terms of the ability to use and of the availability of adequate tools [36, 37]. There is also international agreement that distance learning has further affected students already on the margins of school community, such as SEN students or those with cultural and economic difficulties, increasing social inequalities [21, 38, 39]. For all these reasons, the Italian situation may represent a good case to be studied to understand the effects of long-time school closures. We decided to focus our investigation on primary school level, as it is a crucial period for pupils' learning outcomes.

## Aim of the present study

The current investigation intends to provide an overview of the effects of the school closure due to the COVID-19 pandemic on the learning outcomes of Italian primary school students (target population) according to their opinion, and the opinions of their teachers and/or parents. As shown in the Introduction, the school closure affected not only the academic achievement of students, but also several dimensions. Hence, our work intends to identify all dimensions related to the scholastic well-being invalidated from the school closure due to COVID-19. Moreover, we wanted to investigate if there was agreement or not between the opinion of the students themselves, and their parents and teachers' opinion, as well as set off differences or similarities with international literature. Therefore, our work intends to answer three specific Research Questions:

- Research Questions #1: Beyond academic performance, are there other aspects linked to scholastic well-being of primary school students that were affected by school closures?

- Research Questions #2: Is there agreement between the opinions of parents, teachers, and students regarding the different effects of school closures on Italian primary school students?

- Research Questions #3: Are there any differences between the effects of school closures on Italian primary school students and the students in other countries as appears in the international literature on the topic?

With these aims in mind, we intend to follow a rigorous search strategy to identify all studies focused on the effects of school closures due to COVID-19 on Italian primary school students, based on the point of view of their parents, teachers, and the students themselves. We are aware that there are already numerous reviews on the effects of the pandemic on education. However, we believe that our research has many points of originality, and it has the potential to add new and important information to the literature in the field. First, our systematic review is the only review addressing the pandemic's effect on education on an Italian primary students population. This represents an important adding point to the literature not only because of the difference in nationality per se, but also because the investigated phenomenon impacted Italy, with respect to all the other European countries, in a peculiar and severe manner that made Italy a unique source of investigation of the impact of this phenomenon on education, regardless of the nationality. Thus, we strongly believe that the Italian point of view in this field is not just one other point of view together with the one of other countries, but it has a dramatic privilege to be the point of view of the European country in which children experienced the longest school closures and one of the greatest COVID-19 impact in terms of deaths and infections in the first pandemic months. Telling about the impact of COVID-19 pandemic on the Italian students' population means telling the international literature about the European country in which this tragic event has potentially had the greatest effect.

## Method

### Literature search and search strategy

The articles were searched according to the PRISMA guidelines [40]. The study was preregistered on the Open Science Framework (https://doi.org/10.17605/OSF.IO/P4FGM –March 17th, 2023). On February 2nd, 2023, the first and second authors searched on 4 different databases: Web of Science; Pubmed; EBSCOHost (including PsycArticles, PsycInfo, Psychology and Behavioral Sciences Collection); Scopus. The same researchers examined the preprint servers PsyArXiv, Open Science Framework, and PROSPERO, searching papers, preprints, or preregistered works on the same topics. Limiting the identification to the topic section (title, abstract, keywords), they used the following search string: *(COVID OR CORONA OR "SARS-COV-2") AND (ITAL\*) AND (SCHOOL OR "PRIMARY SCHOOL" OR "ELEMENTARY SCHOOL" OR SCHOLASTIC) AND (LEARN\*)*. Next, we used both the backward (the works cited in the selected articles) and forward (the studies that cited the considered articles) methods to identify any further studies.

### Inclusion and exclusion criteria

We retained any article published between March 9th, 2020 (first COVID-19 lockdown in Italy) and February 2nd, 2023 (article search date); written in English or Italian. Study participants had to be Italian primary school students and/or their teachers/parents, but always with a specific focus on Italian primary school students' leaning during COVID-19 lockdown. We excluded studies that did not satisfy those criteria. Furthermore, we excluded non-journal papers, editorials, dissertations, letters to authors, comments on published articles, and grey literature in general.

### Screening process

Eligible studies were screened using Rayyan [41]. After removing duplicates, the first two authors assessed the titles and abstracts of all identified articles compared to the predefined criteria, discussing any disagreements later with the last two authors. The same researchers

reviewed the text of the selected publications and performed reliability using the Cohen's Kappa (κ) to measure agreement between two reviewers. Cohen's Kappa was interpreted using Landis & Koch [42] convention as Poor (< 0.00), Slight (0.00–0.20), Fair (0.21–0.40), Moderate (0.41–0.60), Substantial (0.61–0.80), and Almost Perfect (0.81–1.00) Agreement.

## Quality assessment

Given the heterogeneity of the studies included, we used different tools to assess the quality of the evidence and the risk of bias. We used the Risk of Bias in Non-randomized Studies–of Interventions tool (ROBINS-I) [43] for the assessment of the studies that used an experimental methodology [44]. All the other studies used an observational methodology, therefore we used the Newcastle-Ottawa Scale (NOS) [45] in three different versions, depending on the type of study: cohort studies (NOS–C); case-control studies (NOS–CC); cross-sectional studies (NOS–CS). Our systematic review includes: 3 cohort studies [46–48]; 1 case-control study [49]. The remaining 29 are cross-sectional studies. The first two authors conducted an independent assessment of study quality, calculating percent agreement and Cohen's K [50]. Any concerns were discussed subsequently with the other authors.

## Results

### Data extraction

380 papers potentially relevant were found: 176 papers from Scopus, 116 from Web of Science, 61 from PubMed, and 27 from EBSCOhost. See Fig 1 for a PRISMA flowchart of the literature search process.

First, we detected and removed 181 duplicates. We screened the titles and abstracts of the remaining 199 articles, removing 146 papers: 82 because of the wrong outcome (no reference to student learning); 63 for the wrong population examined (pre-school, middle-school, high-school, university, or others); 1 for the paper's language (different to English or Italian). Four records were excluded as the full text was not available. The first and second author independently reviewed the remaining 49 eligible studies, according to the criteria defined in advance. This process led to the exclusion of 17 papers: 13 for a different outcome (not focused on Italian primary school students' learning); 3 for the population (different from an Italian sample); 1 for the type of publication (dissertation). Then, we included 2 articles consulting the papers' references via backward and forward search. We computed a Cohen's Kappa (κ) of 0.80 –a substantial agreement level [42]. We discussed emerging discrepancies with the last two authors. We retained 34 articles suitable for inclusion.

### Risk-of-bias assessment

The first two authors independently assessed the methodological quality of the studies with ROBINS-I [43], and NOS [45] in three different versions (cohort, case-control, and cross-sectional studies). When the independent evaluation was finished, the two authors obtained a percentage of agreement of P(a) = 95% and a Cohen's K of 0.64 –a substantial agreement level [50]. All differences were discussed subsequently. Regarding the single study with a non-randomized experimental methodology [44], no concerns and/or high risk of bias were highlighted, and the study was judged to be at low risk of bias for all domains (see S1 Fig). The same was true for the three studies evaluated through the NOS-C [46–48], that obtained a score of 7 (good score level–see S1 Table). All these studies presented an unsatisfactory value only in the Outcome section, due to the lack of information about the method of evaluating the results (e.g., independent blind assessment). Only one study [49] was evaluated through

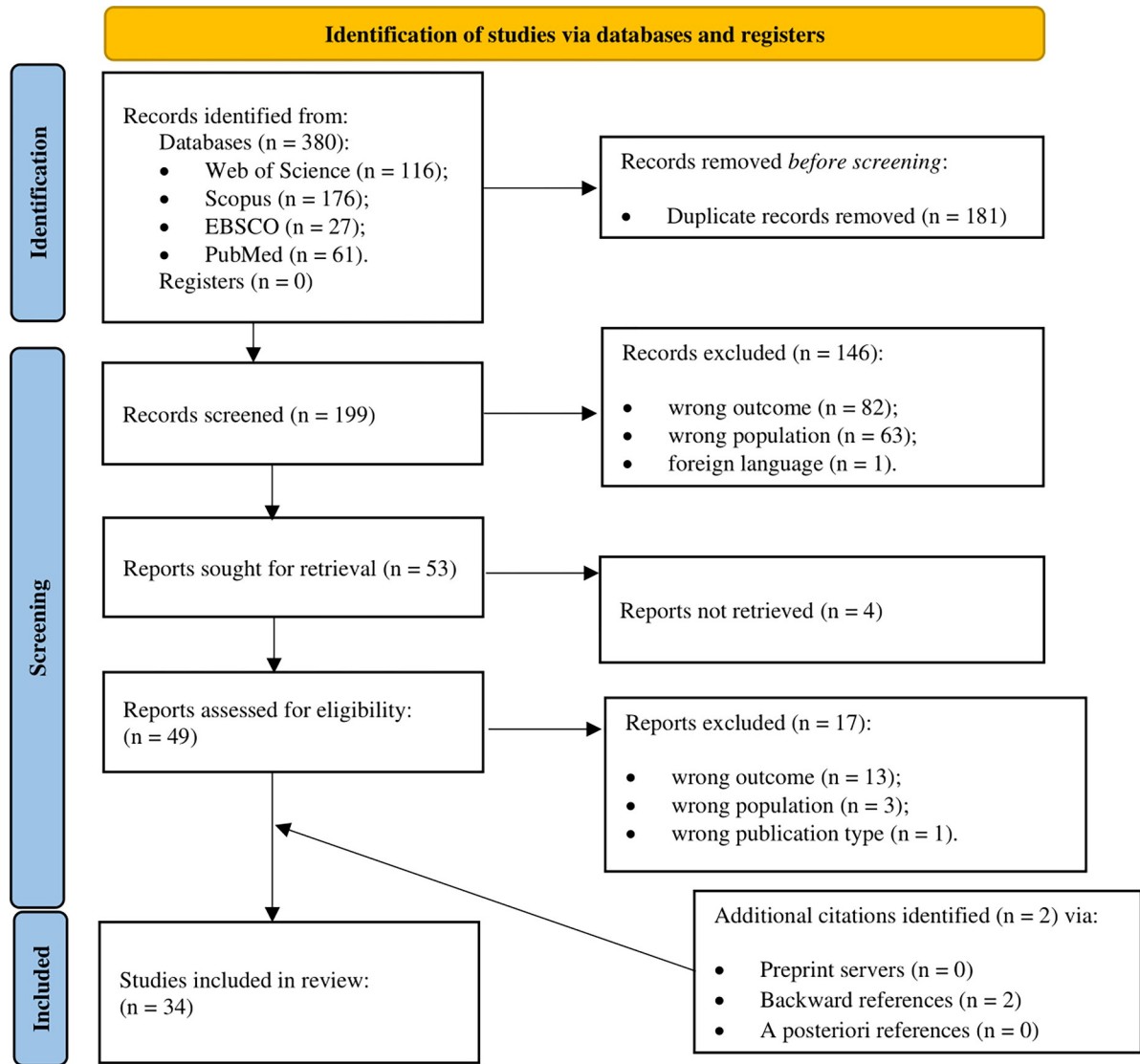

**Fig 1. PRISMA flowchart of the literature search and screening process.**

the NOS-CC, and obtained a total score of 5 (satisfactory–see S2 Table). The unsatisfactory scores are mainly due to the lack of information about the rate of non-responders and the self-report data collection method (survey). 29 studies were evaluated through the NOS-CS (S3 Table). Only one study achieved a good level of evaluation [51] and only 2 a satisfactory criterion [52, 53]. There is still much debate within the scientific community on the inclusion of studies highly vulnerable to bias within systematic reviews and meta-analyses [54, 55]. We decided to maintain studies with high risk of bias for different reasons. First, assessing the risk of bias does not necessarily imply the possibility that a study is biased, but only that it is susceptible to bias [56]. Second, we decided to preserve our research by Tipe II error, thus the possibility to exclude informative papers because of a too stringent methodology impossible to adopt in consideration of the pandemic period and the covered topic. Third, it should be noted that these high risks of bias levels are mainly due to the choice of sampling methods (snowball) and the evaluation methods used (surveys or structured interviews). This choice is

to be considered binding as the pandemic period in Italy consisted in both the closure of research institutions and the impossibility of researchers to conduct experimental studies involving schools.

## An overview of the effects of the school closure due to the COVID-19 pandemic

In addition to the scholastic achievement, several studies included in our systematic review found that the school closure due to Covid-19 affected Italian primary school students on an emotional level (see, for example, Ref. [57]), particularly for those who were already on the margins of the educational system prior to the COVID-19 pandemic, due to socio-cultural, socio-economic difficulties, and/or difficulties due to one or more diagnostic conditions (e.g., Ref. [53]). According to the studies included in our work, some students may have encountered difficulties during distance learning, from a technical-practical point of view, such as: difficulties in following the online lesson, technical problems or competence of teachers/students with the new learning methodology (for example, Ref. [58]). Hence, we categorized the articles according to the specific dimensions investigated, namely: emotional aspects of the learning context; marginality-inequality effects; didactic problems; learning quality. See Table 1 for a synthesis of the studies included and the investigated dimensions.

The section called "Emotional aspects of the learning context" includes studies investigating emotional aspects experienced by students in distance learning during the pandemic period. By the general term "emotional" we mean studies that considered both short-term (e.g., fear, joy, sadness, anger) and long-term (e.g., anxiety, frustration, serenity) emotional states. We also include cognitive processes linked to emotional responses, such as attention and motivation to learn [83, 84]. We divided the studies according to the involved sample's opinion: parents; teachers; students. The study by Crescenza et al. [65] was the only one that investigated these aspects from the perspectives of both teachers and parents. Therefore, we included the study in both sub-categories. In summary, 2 studies collected information from a sample of teachers, 8 from parents, and 1 from students. We repeated the same distinction also in the other sections.

Within the section "Marginality-inequality effects", we included the papers that have at-risk students within their sample: students with specific learning disorders (SLDs), neurodevelopmental disorders (e.g., autism spectrum disorder, ASD; attention deficit and hyperactivity disorder, ADHD), or neurodiverse students in general. We also included students with SEN, linguistic-cultural difficulties, and participants with low SES. Of the 20 papers included, 7 collected information from a sample of parents, 6 from teachers, and 7 from students. However, Baschenis et al. [44] and Inguscio et al. [72] collected data through questionnaires to parents and tested children through psychometric tests; therefore, we placed them in both subsections. Consequently, the section includes 7 papers that collected information from a sample of parents, 6 from teachers, and 9 from students.

The new teaching approach adopted during distance learning has brought difficulties in evaluating student performance, as well as didactic problems encountered by parents or teachers themselves. The category called "Didactic problems" includes studies that considered problems encountered during distance learning, such as the provision of the teaching itself, technical problems, or competence of the teachers/pupils in the use of the new learning tools, issues with the online assessment of students' achievements. This section includes 11 studies that dealt with these issues in the pandemic period: 2 according to the parents' opinion; 9 to the teachers' opinion. No articles evaluated this topic according to students' opinion.

**Table 1. Synthesis of studies included.**

| Authors | Investigated dimension(s) | | | |
| --- | --- | --- | --- | --- |
| | Emotional aspects of the learning context | Marginality-inequality effects | Didactic problems | Learning quality |
| Baschenis et al., 2021 [44] | | ✓ | | |
| Bazoli et al., 2022 [46] | | ✓ | | ✓ |
| Bazzoli et al., 2021 [58] | | ✓ | ✓ | |
| Benigno et al., 2020 [59] | | ✓ | | |
| Bertoletti et al., 2023 [52] | | | ✓ | ✓ |
| Borgonovi & Ferrara, 2023 [47] | | ✓ | | ✓ |
| Canals-Botines et al., 2021 [60] | | | ✓ | |
| Capperucci et al., 2022 [61] | | | ✓ | ✓ |
| Capurso & Roy Boco, 2023 [62] | | ✓ | | |
| Champeaux et al., 2022 [63] | ✓ | | | |
| Colombo & Santagati, 2022 [64] | | ✓ | | |
| Contini et al., 2022 [48] | | ✓ | | ✓ |
| Crescenza et al., 2021 [65] | ✓ | ✓ | | |
| Crisci et al., 2021 [53] | | ✓ | | |
| Decarli et al., 2022 [66] | | ✓ | ✓ | ✓ |
| Doz et al., 2023 [51] | ✓ | | ✓ | |
| Ferretti et al., 2021 [67] | | | ✓ | |
| Gaggi et al., 2020 [68] | | | ✓ | |
| Gentile et al., 2021 [69] | | | | ✓ |
| Guzzo et al., 2022 [70] | | | ✓ | |
| Ianes & Bellacicco, 2020 [71] | | ✓ | | |
| Inguscio et al., 2023 [72] | | ✓ | | |
| Mangiavacchi et al., 2021 [57] | ✓ | | | ✓ |
| Marchese et al., 2022 [73] | | ✓ | | |
| Picca et al., 2021 [74] | ✓ | | | |
| Ranieri et al., 2020 [75] | | ✓ | ✓ | |
| Scarpellini et al., 2021 [76] | ✓ | ✓ | ✓ | ✓ |
| Segre et al., 2021 [77] | ✓ | | | |
| Szpunar et al., 2021 [78] | ✓ | | | |
| Termine et al., 2021 [79] | | ✓ | | |
| Termine et al., 2022 [49] | | ✓ | | |
| Thorell et al., 2021 [80] | ✓ | ✓ | | |
| Thorell et al., 2022 [81] | | ✓ | | |
| Zaccoletti et al., 2020 [82] | ✓ | | | |
| **TOTAL** | **10** | **20** | **11** | **9** |

The category called "Learning quality" includes studies that compared academic performance during the pandemic with pre-pandemic performances. Of the 9 papers included, 3 collected information from a sample of parents, 3 from teachers, and 3 from students.

Therefore, this work includes 10 articles dealing with emotional aspects of the learning context, 20 related to marginality-inequality effects, 11 on didactic problems, 9 on learning quality. Some articles investigated two or more areas of learning; therefore, they were included in each dimension investigated. See Table 2 for sample characteristics and demographic information about the studies included.

**Parents' opinion.** Fourteen studies have delved into the effects of distance learning during the COVID-19 pandemic from the perspective of parents [44, 53, 57, 63, 65, 68, 69, 72, 74, 76,

**Table 2. Descriptive synthesis of the included studies.**

| ID | Authors | Sample | | | | | Italian location (s) | |
|---|---|---|---|---|---|---|---|---|
| | | Type | N | Sex | Age | | | Sd |
| | | | | | Mean | | | |
| 1 | Baschenis et al., 2021 [44] | P/S | 63 parents' children with dyslexia (EG). 38 parents' children without dyslexia (CG) | NA | NA | | NA | North (Pavia, Lombardy) |
| | | | 65 with dyslexia (EG); 52 without dyslexia (CG) | F = 20 with dyslexia | 10.64 with dyslexia | | 1.60 with dyslexia | |
| | | | | F = 20 without dyslexia | 9.80 without dyslexia | | 1.57 without dyslexia | |
| 2 | Bazoli et al., 2022 [46] | S | '18/'19 samples: 24.781 students for 5th grade; 29.675 8th grade. '20/'21 samples: 16.631 for 5th grade; 9.708 8th grade. | NA | NA | | NA | NA |
| 3 | Bazzoli et al., 2021 [58] | T | 3121 (39.4% primary school teachers) | NA | 51 | | NA | 48% North, 22% Center, 30% South* |
| 4 | Benigno et al., 2020 [59] | T | 12 | F = 12 | NA | | NA | 8 children's hospitals in Italy (4 located in North, 1 in Center, and 3 in South) |
| 5 | Bertoletti et al., 2023 [52] | T | 1407 (51% primary school teachers) | F = 93% | 48.94 | | 0.64 | 52% from North |
| 6 | Borgonovi & Ferrara, 2023 [47] | S | <1mn (985.909 primary school students) | NA | NA | | NA | NA |
| 7 | Canals-Botines et al., 2021 [60] | T | 31 (Italian sample) | (Italian sample) F = 29 M = 2 | NA | | NA | Center (Lazio) |
| 8 | Capperucci et al., 2022 [61] | T | 11.828 (47.5% primary school teachers) | NA | NA | | NA | Center (Tuscany and Umbria) |
| 9 | Capurso & Roy Boco, 2023 [62] | T | 21 (10 primary school teachers) | F = 90% | 43.6 | | 8.8 | NA |
| 10 | Champeaux et al., 2022 [63] | P | 2.455 Italian and 1.838 French families (3.769 Italian and 3.183 French children. The 47.1% of Italian children were primary school students) | NA | Italian children F +M = 8.3 (between 2–18 years old) | | Italian Children F+M = 3.5 | 54.9% North, 25.7% Center, 19.4% South* |
| 11 | Colombo & Santagati, 2022 [64] | T | 145 (53% Italian primary school students) | F = 95% of the entire sample | NA | | NA | For the entire sample: 50% North, 15% Center, 35% South |
| 12 | Contini et al., 2022 [48] | S | 1539 | NA | NA | | NA | North (Torino) |
| 13 | Crescenza et al., 2021 [65] | P/T | 991 parents (57% parents' primary school students) | F = 826 (83.4%) | NA | | NA | North, Center, South (Lombardy, Lazio, Puglia) |
| | | | 794 teachers (39% primary school teachers) | F = 674 (84.9%) | NA | | NA | |
| 14 | Crisci et al., 2021 [53] | P | 637 parents (339 primary school students) | Parents F = 92% | NA | | NA | NA |
| | | | | Children M = 48% | Children F+M = 10.8 | | Children F +M = 3.24 | |
| 15 | Decarli et al., 2022 [66] | T | 120 (59 primary school teachers) | F = 105 | 49.66 | | 9.5 | For the entire sample: 14 schools from North, 10 from Center, 21 from South |
| 16 | Doz et al., 2023 [51] | T | 270 (135 of primary schools) | F = 239 | 49.5 | | 9.74 | NA |

(*Continued*)

**Table 2.** (*Continued*)

| ID | Authors | Sample | | | | | Italian location (s) | |
|----|---------|--------|--|--|--|--|------|--|
| | | Type | N | Sex | Age | | | |
| | | | | | Mean | Sd | | Sd |
| 17 | Ferretti et al., 2021 [67] | T | 244 (122 primary school teachers) | NA | NA | | NA | NA |
| 18 | Gaggi et al., 2020 [68] | P | 490 families | NA | NA | | NA | 17 out of 20 Italian regions |
| 19 | Gentile et al., 2021 [69] | P | 19.527 families (almost 40% of 32.000 children were from primary schools) | F = 86% (the responding was generally the mother) | 30.8% of them were 45–49 years old. | | NA | 68.9% from Center |
| 20 | Guzzo et al., 2022 [70] | T | 977 (315 primary school teachers) | F = 792 (81%) | Less than 28 (1%)– more than 60 (15%). 50–59 (44%) | | NA | 48% North, 12% Center, 40% South* |
| 21 | Ianes & Bellacicco, 2020 [71] | T | 3.291 (40,9% primary school students) | NA | NA | | NA | For the entire sample: 42.3% North, 15.2% Center, 42.4% South |
| 22 | Inguscio et al., 2023 [72] | P/S | 61 parents | F = 51 (mothers– 83.6%) | F = 44.09 M = 49.66 | F = ±6.23 M = ±4.69 | Center (Polyclinic Umberto I, Rome) | |
| | | | 61 children | F = 36 (daughters– 59.02%) | F = 12.19 M = 12.33 | F = ±2.96 M = ±3.01 | | |
| 23 | Mangiavacchi et al., 2021 [57] | P | 2101 families (3619 Children under 16, 40.3% primary school students) | Parents F = 93% of respondents were mothers | Parents F = 41 M = 44 | | NA | 53.6% North, 26.7% Center, 19.7% South |
| | | | | Children M = 50.7% | Children F+M = 7 | | NA | |
| 24 | Marchese et al., 2022 [73] | S | 33 (15 primary school students) | Primary school students F = 6 | Primary school students = 7 | | NA | South (Barletta) |
| 25 | Picca et al., 2021 [74] | P | Parents of 3392 children [1704 (50.2%) Italian primary school students] | Italian primary school students M = 53% | Italian primary school students: between 6 and 10 | | NA | NA |
| 26 | Ranieri et al., 2020 [75] | S/T | 820 | NA | NA | | NA | Different Italian regions |
| 27 | Scarpellini et al., 2021 [76] | P | F = 1.601 mothers (1.138 mothers of primary school students) | Mothers' primary school students F = All | Mothers' primary school students F = 42,5 | | NA | 70.2% from North |
| | | | Primary school students F+M = 1.148 | Primary school students F = 545 (48.2%) | Primary school students F+M = 8 | | NA | |
| 28 | Segre et al., 2021 [77] | S | 82 (54.9% primary school students) | F = 46.3% | 10.4 | | NA | North Italy (Milan) |
| 29 | Szpunar et al., 2021 [78] | P | 5022 parents | Parents F = 4452 | Parents = 43.51 | Parents = 5.99 | For the entire sample: 25.3% North, 64.2% Center, 10.5% South* | |
| | | | 5822 children (38% primary school students) | Children = F = 2830 | Children = 9.3 | Children = 4 | | |
| 30 | Termine et al., 2021 [79] | S | 1362 with Neurodevelopment disorders (NDD– 455 primary school students) | M = 861 with NDD | primary school students = 9 | | NA | North (Varese) |
| | | | 6943 without NDD (3318 primary school students) | M = 3280 without NDD | | | | |
| 31 | Termine et al., 2022 [49] | S | with tic disorder = 49 | M = 39 | 6–18 years | | NA | North (Varese) |
| | | | Control group = 245 | M = 125 | NA | | | |
| 32 | Thorell et al., 2021 [80] | P | 6720 parents (794 from Italy) | NA | NA | | NA | NA |
| 33 | Thorell et al., 2022 [81] | P | 1010 from different European Countries | NA | NA | | NA | NA |

(*Continued*)

**Table 2.** (Continued)

| ID | Authors | Sample | | | | | Italian location (s) | |
|---|---|---|---|---|---|---|---|---|
| | | Type | N | Sex | Age | | | |
| | | | | | Mean | | | Sd |
| 34 | Zaccoletti et al., 2020 [82] | P | 567 (Italian's parents = 173) | F = 89% (Italian's mother) | 42,90 (Italian's parents) | | 5,76 (Italian's parents) | NA |
| | | | Italian primary school students F +M = 72% | F = 46% (Italian's children) | 9,65 (Italian's children) | | 2,14 (Italian's children) | |

P = Parents. S = Students. T = Teachers. M = Male. F = Female. NA = Not available.

*The southern area marked with an asterisk includes the major islands (Sicily and Sardinia).

78, 80–82]. Online questionnaires were the main method of data collection (see Table 3 for a description of the studies included in this section). Consensus among these studies is that distance learning has posed significant challenges for children. Concerns about the quality of learning during distance education were widespread among parents, with many expressing dissatisfactions with the effectiveness of online instruction [57, 69, 76]. Parents have reported difficulties in supporting their children academically [65], leading to disruptions in children's emotional well-being, including increased aggression and restlessness [63, 76]. There has also been a noticeable decrease in attention, concentration, and engagement during home study sessions [80, 82]. Additionally, concerns have been raised about increased screen time and decreased participation in school and extracurricular activities [57, 78, 82]. Despite these challenges, one study found a positive effect on children's sleep and family relationships due to parents' remote work [74]. Furthermore, several parents reported amplified difficulties faced by marginalized and neurodiverse students during distance learning. Regarding to the marginality-inequality condition, parents of students with special educational needs (SEN) or neurodevelopmental disorders have expressed dissatisfaction with the support received and highlighted the negative impact on their children's academic performance and psychological well-being [53, 80, 81], along with higher levels of fidgetiness [65]. Similarly, parents of dyslexic children have reported significant difficulties in online learning compared to parents of neurotypical children [44]. Inguscio et al. [72] investigated the perception of online learning by students (and their parents) with or without hearing loss (HL). The presence or absence of HL did not influence the parents' estimation of the difficulties with the distance learning. Over 50% of the parental population defined the distance learning as quite useful, although with lower scores than the direct relationship (in class) between child-teacher and schoolmates. Furthermore, instructional problems have been noted, with some children unable to participate in distance learning due to technological constraints or perceived inefficacy of remote learning methods [68, 76]. The investigation conducted by Gaggi et al. [68] found that 4.5% of the children in their sample (22 out of 490) never participated in distance learning during the pandemic, due to family technological difficulties or school impreparation, or even because some teachers considered it a suboptimal method for teaching young students. More than 70% of the parental sample of Scarpellini et al. [76] reported that they were not in favor of the learning methods imposed by the distance learning, and many of them (80%) found themselves in great difficulty in giving technological help to children.

**Teachers' opinion.** Fourteen studies have investigated teachers' opinions on emotional aspects, marginality-inequality effects, instructional issues, and the quality of their students' learning [51, 52, 58–62, 64–67, 70, 71, 75]. Only two studies investigated emotional status of students according to their teachers [51, 65]. The teachers' sample by Doz et al. [51] observed a

**Table 3. Parents' opinion.**

| Authors | Aim | Assessment tool | Investigated dimension(s) | Results |
|---|---|---|---|---|
| Baschenis et al., 2021 [44] | A cross-sectional comparison of reading skills between children with and without dyslexia evaluated pre- and post-lockdown | For parents, an ad hoc questionnaire. EG was assessed before and after the lockdown; CG only after. | Marginality-inequality effects | None of the CG parents reported difficulties in their children following online lessons, unlike the EG parents, who expressed difficulties in carrying out their homework. Both groups of parents reported worsening scholastically and relationally. |
| Champeaux et al., 2022 [63] | Evaluation of Italian/French parents on children's home-schooling process and emotional well-being | Online questionnaire | Emotional aspects of the learning context | For the Italian sample, there was a loss of learning (M = −5.138; SD = 2.614) and a worsening of the emotional well-being (M = −0.670; SD = 0.957) of their children |
| Crescenza et al., 2021 [65] | Compare the opinion of all participants involved in school learning on the distance learning | Online questionnaire | Emotional aspects of the learning context | Parents report having had to apply several times for a work permit to support their children during the distance learning. Increased levels of restlessness among students with disabilities. |
| | | | Marginality-inequality effects | Increased levels of restlessness among students with disabilities. |
| Crisci et al., 2021 [53] | Investigate the role of background, child (executive functions), and parental (psychological well-being) factors in the perception of negative/positive effects of distance learning | Online questionnaire, based on: Childhood Executive Functioning Inventory (CHEXI); Positive Mental Health (PMH) | Marginality-inequality effects | The negative effects of distance learning were related to a younger age and a greater EF deficit. Parents' psychological well-being had no significant effect |
| Gaggi et al., 2020 [68] | Evaluate the technologies and methodologies adopted during the COVID-19 period by teachers and the impact it has had on students and their families | Online questionnaire | Didactic Problems | Regarding to learning, 22 families (4.5%) reported that the school did not activate distance learning at all, to be understood as any type of activity related to learning, from passive activities, such as sending/checking homework, to the same distance learning. |
| Gentile et al., 2021 [69] | Evaluate the technologies and methodologies adopted during the COVID-19 period | Online questionnaire | Learning Quality | Regarding to distance learning's efficacy, parents' primary school students reported, in a 10-point Likert, an average of 5 point. |
| Inguscio et al., 2023 [72] | Investigate the psychological characteristics of distance learning on Italian students with/out HL | Online questionnaire, based on: Questionnaire on school wellbeing (QBS); State-Trait Anxiety Inventory for adults (STAY-Y); Revised Children's Manifest Scale (RCMAS-2) | Marginality-inequality effects | No significant difference of anxiety between parents, reported as "not particularly problematic" according to the Italian norms. |
| Mangiavacchi et al., 2021 [57] | Investigate the parents' reactions to the lockdown and the role of these changes on children's emotional wellbeing, time with their parents and their educational outcomes | Online questionnaire | Emotional aspects of the learning context | Parents observed a significant reduction in their children's emotional wellbeing. |
| | | | Learning Quality | Parents observed a reduction in productive activities (school, homework, extracurricular activities): from around 60% of daily activity to less than 40%. Slight improvement in reading time. |
| Picca et al., 2021 [74] | Investigate the positive/negative effect of COVID-19 on children's behaviors, daily life, distance learning, and the use of digital device | Online questionnaire | Emotional aspects of the learning context | For the Italian primary school students, the improvement of family relationships and parents' remote working positively influenced sleep, emotional and behavioral disturbances; the same were negatively influenced by screen time and time spent watching TV |

(*Continued*)

**Table 3.** (Continued)

| Authors | Aim | Assessment tool | Investigated dimension(s) | Results |
|---|---|---|---|---|
| Scarpellini et al., 2021 [76] | Evaluation of mothers on children's home-schooling process and emotional well-being | Online questionnaire | Emotional aspects of the learning context | Mothers observed behavioral changes in their children, specifically restlessness (69.1%) and aggressiveness (33.3%) |
| | | | Marginality-inequality effects | Over the 50% of the children with not specified chronic disorders attended distance learning only once a week. However, almost the 15% of them did not receive any specific support. |
| | | | Didactic Problems | Although most experienced no difficulties with the technology (80.7%), mothers did not approve distance learning (72.2%). |
| | | | Learning Quality | 11.5% of elementary school students were either not evaluated, didn't obtain any grades (53.1%), or received lower grades compared to their prior academic performance (7.1%). Nevertheless, a majority of primary school students assessed, specifically 80%, received identical grades as they did before the COVID-19 pandemic. |
| Szpunar et al., 2021 [78] | Investigate the changes in students' lifestyle habits due to the pandemic, in particular the school activities. | Online questionnaire | Emotional aspects of the learning context | 12.6% of primary school students spend more than 3 hours per day doing homework, more time than daily distance learning. The 56.1% of the same students do homework more than before of the lockdown (20.3% less) |
| Thorell et al., 2021 [80] | Comparison of parenting experiences on homeschooling from different European countries | Online questionnaire | Emotional aspects of the learning context | The results referring to Italy highlight a strong parental concern for the negative effects of the distance learning on their children. Also, not everyone has been able to stick with homeschooling fluently |
| | | | Marginality-inequality effects | 49.1% of parents of SEN students reported that the support they received during distance learning was not sufficient. |
| Thorell et al., 2022 [81] | Investigate the parents' perception about the role of ADHD/ASD's deficit or both on the positive/negative effects of distance learning | Online questionnaire | Marginality-inequality effects | Higher levels of negative effects of distance learning in children with ADHD and/or ASD. |
| Zaccoletti et al., 2020 [82] | Examine the opinion of Italian and Portuguese parents about impact of lockdown on students' academic motivation and their participation in extracurricular activities | Online questionnaire, based on AMOS 8–15 | Emotional aspects of the learning context | Decreased academic motivation and extracurricular activities in both Countries |

decline in students' attention and motivation, particularly among those lacking parental support. Over 60% of the surveyed teachers by Crescenza et al. [65] acknowledged the importance of prioritizing the emotional well-being of students and families. However, they also expressed concerns about their perceived lack of proficiency in fulfilling this role. It is indisputable that the pandemic has had disruptive effects on student populations who already presented a difficult clinical-family picture; within the school context there was inevitably a loss or a slowdown in learning [58, 59]. Teachers reported that their students with special educational needs (SEN) have faced significant challenges, including changes in relationships, decreased autonomy, and passive participation [62, 66], or even no online lesson activities for these students,

given the difficulties in delivering their personalized teaching plans online [64, 71]. While some teachers recognized the benefits of distance learning, such as innovative teaching methods, many encountered difficulties in building relationships with students and adapting to technological limitations [52, 70]. As regards didactic problems, digital competence has emerged as a critical issue, with widespread resistance to digital methods among older teachers [51, 52, 60, 66]. This resistance has negatively influenced student performance, underscoring the importance of ongoing training and support for teachers [70]. Additionally, assessing students' academic performance in a remote environment has posed significant challenges, necessitating adaptation of assessment methods. Capperucci et al. [61] reported that the most common methods in primary school, during the pandemic period, were homework (78.7%), tests (63.3%), oral questions (62.0%), unspecified exercises (59.0%). Over 60% of teachers who responded to the survey by Ferretti et al. [67] did not believe assessment was feasible in distance learning environments. See Table 4 for a description of the studies on teachers' opinion.

**Students' opinion.**   As highlighted in the preceding sections, students have faced various emotional challenges and learning difficulties due to schools closure during the pandemic period. This section provides a summary of the results of 10 studies that investigated students' own opinions [44, 46–49, 72, 73, 75, 77, 79]. See Table 5 for a description of every study included in this section.

Segre et al. [77] conducted interviews revealing increased levels of anxiety (over 75%) and concerns about family members contracting COVID-19 among almost 80% of students in their sample. Dyslexic students faced heightened difficulties with online lessons and homework compared to their peers without specific learning disorders [44], although such difficulties in following online lessons were not highlighted by the sample of SLDs students investigated in Marchese et al. [73]. However, their sample (21 students from a single primary school, in southern Italy) cannot be considered representative. Inguscio et al. [72] did not find a significant impact of hearing impairment on academic well-being, while Contini et al. and Ranieri et al. [48, 75] highlighted exacerbated educational disparities, particularly among socioeconomically disadvantaged students. The pandemic has worsened existing disparities among students, especially those from lower socioeconomic backgrounds, as Borgonovi & Ferrara [47] demonstrated a non-linear relationship between the pandemic's effects on learning and Economic, Social, and Cultural Status (ESCS) [85], with medium-low SES students experiencing the greatest setbacks. Termine et al. [49, 79] also noted significant challenges for students with neurodevelopmental disorders (NDDs), attributing the difficulties to the inadequacy of distance learning for their individualized educational needs. Termine et al. [49] collected information on a sample of 49 patients between 6 and 18 years old affected by tic disorders (common hyperkinetic movement disorder in childhood). More than 50% of their sample reported an increase in the severity of tics during the pandemic, with an increase in restlessness in 30% of the participants and in perceived pain in 21% of them. According to the regression analyses carried out by Termine et al. [49], the persistence and increase of tics was directly associated with a learning difficulty in the distance learning modality. In 2021, Termine et al. [79] conducted a study also in a hospital setting, but with a larger sample size, including patients with tic disorders (n = 46) and not only: 781 with SLD, 125 with ADHD, 42 with ASD. The analyses carried out by Termine et al. [79] reported a strongly negative effects of distance learning condition in students with Neurodevelopmental Disorders (NDD), adapting less than the control group at this new condition for several reason: distance learning difficulties were more pronounced for children with NDD; students with NDD felt more often reprimanded by their parents during the COVID-19 lockdown; they tended to feel less the absence of their peers, indicating a negative impact on the socialization.

**Table 4. Teachers' opinion.**

| Authors | Aim | Assessment tool | Investigated dimension(s) | Results |
|---|---|---|---|---|
| Bazzoli et al., 2021 [58] | Teachers' perspective on distance learning's criticalities and territorial differences during the pandemic | Online questionnaire | Marginality-inequality effects | With greater difficulties in southern Italy, the teachers reported that: 7.7% of the students did not participate in distance learning; 10.3% with irregular participation; 21.8%, although participating regularly, had learning difficulties. |
| | | | Didactic Problems | 58.1% of teachers expressed difficulties in assessing learning during the distance learning. |
| Benigno et al., 2020 [59] | Evaluate the impact of the pandemic on the italian SiHo services | Group interview | Marginality-inequality effects | Hospital teachers report the same difficulties as traditional schools, both from a relational point of view with the students and from an organization of the distance learning |
| Bertoletti et al., 2023 [52] | Identify subgroups of teachers based on the use of digital tools and how these subgroups differ in terms of satisfaction and students' performance. | Online questionnaire + students' standardized test scores (INVALSI) | Didactic Problems | A third of teachers proved resistant to digital methods, using available technologies in a rather limited and resistant way. |
| | | | Learning Quality | The teachers who had greater digital skills were more satisfied with their teaching, positively also associated with better student performance. |
| Canals-Botines et al., 2021 [60] | Spanish, Italian, Romanian teachers' experiences and perspective on distance learning before, during, and after COVID-19 | Online questionnaire | Didactic Problems | Teachers were somewhat prepared to distance learning, although they had struggled to track children's progress and maintain grounding during school closures. |
| Capperucci et al., 2022 [61] | Comparison of national data with data from Tuscany and Umbria on teaching and assessment strategies used during the distance learning | Online questionnaire | Didactic Problems | Teachers who applied alternative methods of assessment provided a higher rating of the students' learning |
| | | | Learning Quality | Regarding the effectiveness of the distance learning for students learning, the average value of the Tuscan sample was lower than the national average, while the Umbrian sample was in line with that. |
| Capurso & Roy Boco, 2023 [62] | Evaluate the changes in roles, relationships and activities caused by school closures in SEN students, according to their teachers. | Semi-structured interviews (56min on average) | Marginality-inequality effects | Several teachers reported that, during the lockdown, SEN students lost the opportunity to relate to the entire class group, often stuck in a dual situation with the support teacher |
| Colombo & Santagati, 2022 [64] | Teachers' experiences and opinions about the strategies used during lockdown with students with disabilities (SD). Furthermore, their emotions on their work, and the relationship with students' parents | Online questionnaire | Marginality-inequality effects | Increased of inadequacy and stress of the majority of teachers (65%, 62% respectively). Most of them reported the difficult to work alone at home (53%). According to the survey responds, 27% of primary school teachers did no activity with SD in their class. Over 40% of teachers did not maintain direct contact with the parents of the pupils. |
| Crescenza et al., 2021 [65] | Compare the opinion of all participants involved in school learning on the distance learning | Online questionnaire | Emotional aspects of the learning context | Most teachers report the need to give more voice to students' emotions. |
| Decarli et al., 2022 [66] | Evaluate the teachers' experience during distance learning and which type of strategies they used with SEN students | Online questionnaire | Marginality-inequality effects | Regarding SEN students, less involvement and greater attention difficulties were highlighted on their part, although teachers reported simplifying the lesson and/or individualizing it for their needs |
| | | | Didactic Problems | Most of the teachers reported feeling comfortable and sufficiently proficient with the technologies used during the distance learning. |
| | | | Learning Quality | They highlighted a disagreement that distance learning helped students learn more efficiently. |

*(Continued)*

**Table 4.** (Continued)

| Authors | Aim | Assessment tool | Investigated dimension(s) | Results |
|---|---|---|---|---|
| Doz et al., 2023 [51] | Investigate the impact of the distance learning on teachers and their views on students' learning difficulties | Online questionnaire | Emotional aspects of the learning context | The teachers reported students' problems in social relationships (33.3%), difficulty in respecting independence (17.4%), motivation (14, 4%), attention (11.8%), and problems with adherence routines (7%). For primary school, a significantly higher number of students relational difficulties (66%). |
| | | | Didactic Problems | The teachers reported students' problems in the use of technologies (53.7%), lack of support from families (7%) and problems with adherence routines (7%). |
| Ferretti et al., 2021 [67] | Evaluate teachers' relationship to new distance learning practices, with a main focus on remote evaluation methods | Online questionnaire | Didactic Problems | Most teachers have adapted to the new teaching methodologies; however, they consider that it is not possible to adequately evaluate in a remote way. |
| Guzzo et al., 2022 [70] | Evaluation of teachers' distance learning experience | Online questionnaire | Didactic Problems | A substantial difficulty in using the new method of delivering lessons was highlighted, trying to involve students as much as possible also through SMS and/or WhatsApp |
| Ianes & Bellacicco, 2020 [71] | One month after the pandemic, evaluate the activation of distance learning for all students and the inclusion of students with disabilities. | Online questionnaire | Marginality-inequality effects | One month after the pandemic, 11.20% of primary school teachers report that they have not activated the distance learning in their class or in some, especially in the south (10.00%). Still regarding primary school, 24.60% of students with disabilities were excluded from the distance learning due to its ineffectiveness, while 9.90% were excluded due to the objectives of the PEI, for which didactic interventions not in presence. |
| Ranieri et al., 2020 [75] | Explore teachers' response to distance learning challenges | Online questionnaire | Didactic Problems | Teachers used different methodologies to allow students to be assessed during the pandemic period, including observation of attitudes, and teamwork. |

It is interesting to note that no studies have been found that highlight whether Italian primary school students experienced difficulties in using the new distance learning methodology, or any technical didactic problems (e.g.: lack of adequate devices, quality of connection Internet). Regarding the quality of learning, three studies used INVALSI data, revealing adverse effects on academic performance, particularly in mathematics, among primary school students during the pandemic. Borgonovi & Ferrara and Bazoli et al. [46, 47] observed a decline in mathematical skills, while reading scores varied across regions. In fact, the results by Borgonovi & Ferrara [47] were in line with those by Bazoli et al. [48], except for the distribution of learning loss on the Italian territory. While Bazoli et al. [46] divided the area of residence into regions, identifying a territorial homogeneity with respect to learning loss, Borgonovi & Ferrara [47] divided the area into Italian provinces (thus a smaller division compared to Bazoli et al.), identifying a somewhat heterogeneous condition. Regarding mathematics, there was more negative results in the provinces further north-west, and in the regions further south (Puglia and Basilicata, and a part of Sicily) compared to the entire national territory. For reading, the areas most affected by learning loss were the north and central north of Italy and the islands. Contini et al. [48] also noted greater losses for students with pre-existing academic difficulties, low socioeconomic status, and female students, especially those with highly educated parents.

**Table 5. Students' opinion.**

| Authors | Aim | Assessment tool | Investigated dimension(s) | Results |
|---|---|---|---|---|
| Baschenis et al., 2021 [44] | A cross-sectional comparison of reading skills between children with and without dyslexia evaluated pre- and post-lockdown | Neuropsychological tests (DDE-2; MT-3-Clinic); ad hoc questionnaire (for children and parents). EG was assessed before and after the lockdown; CG only after. | Marginality-inequality effects | Progressive worsening of reading skills (speed and accuracy) and increase in errors of text, words, non-words. EG's 59%-63% did not reach the expected improvement in reading speed |
| Bazoli et al., 2022 [46] | Investigate the learning loss in math and reading achievement and assess whether educational inequalities are related to the social position of students' families, their geographical area of residence, their migrant status and their educational background | INVALSI ('18-'19 –'20-'21) | Marginality-inequality effects | There is no evidence that COVID-19 worsened the achievements of students in the lower SES more than those of upper-middle SES. |
| | | | Learning Quality | Except for reading among 5th grade students, the pandemic generated learning losses among the attendants of every school grade in both participants tested by INVALSI. |
| Borgonovi & Ferrara, 2023 [47] | Investigate the difference in math and reading achievement of students in '20-'21 (COVID-19) and '18-'19 (non-COVID-19) | INVALSI ('18-'19 –'20-'21) + data from the 2017–2018 and 2015–2016 tests | Marginality-inequality effects | Primary school students belonging to the lowest ESCS quartile (first quartile) achieved better reading scores and lower loss in mathematics in the COVID-19 compared to the non-COVID-19 sample. Those belonging to the second quartile reported the lowest reading gain and the greatest information loss in mathematics. |
| | | | Learning Quality | For primary school students, they showed a smaller reduction in math achievement and an increase in reading achievement in the COVID-19 cohort |
| Contini et al., 2022 [48] | Estimate the effect of school closure on math skills in a primary school sample | INVALSI data: they compared the standardized tests in math and Italian at the end of grade 2 (pre-test) with a standardized assessment administered by the researchers at the end of grade 3 (post-test) | Marginality-inequality effects | The pandemic exacerbated the pre-existing disparities among various socio-economic groups. |
| | | | Learning Quality | Average loss of 0.19 SD in test scores |
| Inguscio et al., 2023 [72] | Investigate the psychological characteristics of distance learning on Italian students with/out HL | Online questionnaire, based on: Questionnaire on school wellbeing (QBS); State-Trait Anxiety Inventory for adults (STAY-Y); Revised Children's Manifest Scale (RCMAS-2) | Marginality-inequality effects | The HL factors did not affect the school wellbeing. They did not find significant difference between children with/out HL |
| Marchese et al., 2022 [73] | Investigate if 1:1 distance learning with a special education teacher was effective in helping SEN children during the pandemic | Online questionnaire | Marginality-inequality effects | Although with difficulties in pursuing and achieving outcomes, all SEN children who followed the platform 1:1 achieved the expected outcomes for their deficit |
| Ranieri et al., 2020 [75] | Explore teachers' response to distance learning challenges | Online questionnaire | Marginality-inequality effects | Over 50% of students did not have proper technological access, due to the family unit not having the necessary devices. Furthermore, the lack of technical (44%) or linguistic skills (33%) on the part of the parents had a particular impact. |
| Segre et al., 2021 [77] | Investigate the students' psychological distress and changes in routine | video-interviews | Emotional aspects of the learning context | 79.3% of the sample reported difficulty adapting to distance learning, perceiving greater learning difficulty at home. Changes in eating and sleeping habits were also highlighted |
| Termine et al., 2021 [79] | Evaluate the impact of social distancing and lifestyle changes on students with/out NDDs during COVID-19 | Online questionnaire | Marginality-inequality effects | Major disadvantage in distance learning for children from low SES families. Worse learning outcomes during distance learning by NDD students |

*(Continued)*

**Table 5.** (Continued)

| Authors | Aim | Assessment tool | Investigated dimension(s) | Results |
|---------|-----|-----------------|---------------------------|---------|
| Termine et al., 2022 [49] | Investigate the impact of the pandemic on patients with tic disorders. | Online questionnaire | Marginality-inequality effects | The presence of tics was associated with greater difficulty with distance learning |

## Discussion

The COVID-19 pandemic has had a considerable impact on learning, generating changes in many psychosocial characteristics of the learning context experienced by children. This systematic review would like to offer an overview of the effects of COVID-19 school closures on Italian primary school students. We did not focus exclusively on the loss in terms of academic performance, but also on the fundamental role that the learning context may have had on the emotions of pupils, according to them and their caregivers. Overall, the findings indicate that younger students struggled with learning outcomes during the pandemic, with a consensus among parents and teachers about the negative impact of distance learning on effective learning, especially in subjects such as mathematics, as demonstrated by the INVALSI measurements [46, 47]. The drastic change in the learning setting was above all studded with technical-practical difficulties, which contributed to increasing inequalities between students, disrupting the learning experience of students who were already considered at risk of marginalization. These difficulties include limited access to devices and reliable internet connection, lack of technology skills among students and teachers, and difficulty managing distance learning effectively.

Following PRISMA guidelines, 34 primary studies were selected that deal with one or more aspects of the learning environment affected by the COVID-19 lockdown. We identified 11 articles dealing with emotions and affective aspects, 20 relating to students most at risk (e.g., specific learning disorders; special education needs; children with linguistic-cultural difficulties), 11 related to educational problems (including student assessment), and 9 about the quality of learning. We considered the opinion not only of pupils, but also of their parents and teachers for each dimension of learning investigated. Fig 2 shows a breakdown of samples by individual dimensions. It should be noted that some articles have dealt with two or more different reference samples.

To our knowledge, this is the first work that considers the opinion of all the relevant figures in school learning with a view that covers not only the individual academic performance, but also the emotional aspects, technological difficulties, and personal problems of Italian primary school students. We consider the identification of these aspects relevant because of several reasons: (1) the emblematic territory (Italy experienced one of the longest periods of school closure due to the pandemic) and sample (primary school students experienced the first years of education in an unprecedented way for the Italian territory); (2) the possibility of future disruptions to the regular school schedule for other exceptional events (e.g., the recent flooding in Emilia-Romagna); (3) the potential contribution to a deeper understanding of the nuanced impact of distance learning on primary school education; (4) the understanding of any consequences of distance learning not only from an observational point of view, but also to implement improvements on a national level, developing more targeted interventions and support systems for those students who may have struggled the most in this new learning environment. Also, by including studies that explore the opinion of teachers, parents and students themselves, this systematic review constitutes the opportunity to observe the same phenomenon by different perspectives which is a rarity in the field.

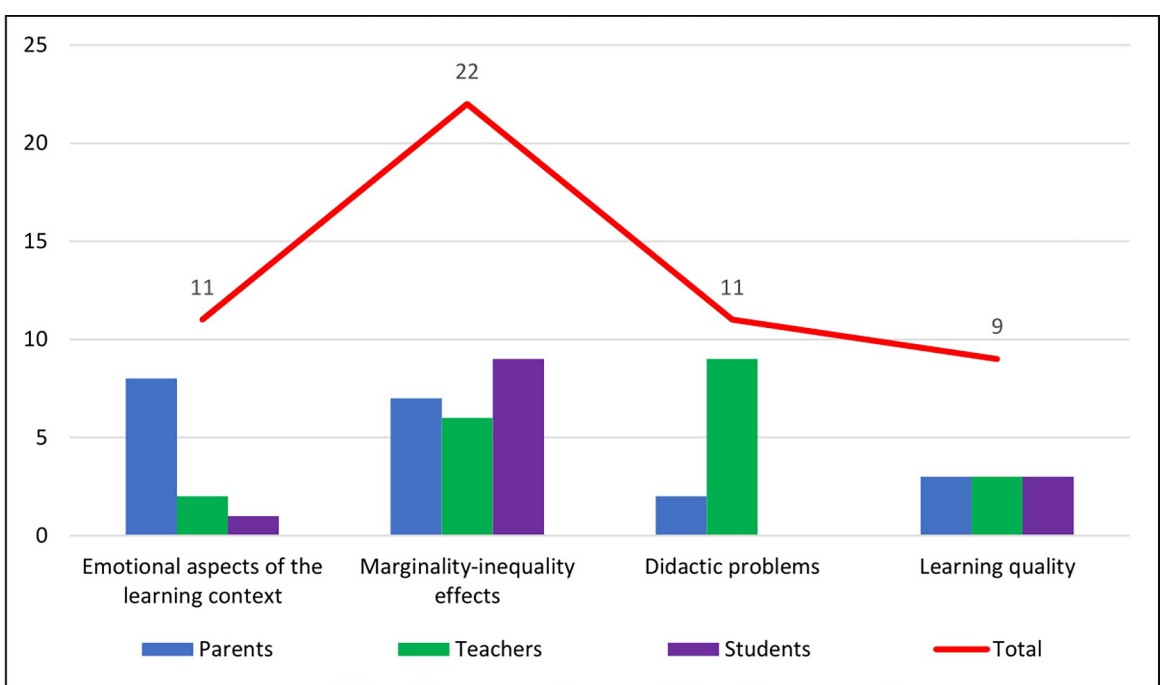

**Fig 2. Components of samples covered by the documents.** X-axis = investigated dimensions. Y-axis = sample size.

### Research Questions #1: Beyond academic performance, are there other aspects linked to scholastic well-being of primary school students that were affected by school closures?

Our systematic review found that, in addition to the scholastic achievement, the school closure due to Covid-19 affected Italian primary school students on an emotional level (see, for example, Ref [57]), particularly for those who were already on the margins of the educational system prior to the COVID-19 pandemic, due to socio-cultural, socio-economic difficulties, and/or difficulties due to one or more diagnostic conditions (e.g., Ref [53]). According to the studies included in our work, some students may have encountered difficulties during distance learning, from a technical-practical point of view, such as: difficulties in following the online lesson, technical problems or competence of teachers/students with the new learning methodology (for example, Ref [58]). In order to achieve an overview of the effects of school closures due to COVID-19 on Italian primary school students, we focused on an analysis of the affected dimensions, according to the studies identified and included in our systematic review. Hence, we subdivided the articles according to the specific dimensions investigated, namely: emotional aspects of the learning context; marginality-inequality effects; didactic problems; learning quality.

**Emotional aspects of the learning context.** Concerning the emotional aspects, most of the articles focused on parental perception. According to these studies, the primary concerns were adequately supporting students in their studies [65] and the potential effects of prolonged interaction with an electronic device [74, 76, 78, 80]. Additionally, the research highlighted emotional alterations, such as disruptive and aggressive mood changes in children [63, 76] and loss of motivation [82]. Compared to the times dedicated to different activities, a heterogeneous condition emerges. While Mangiavacchi et al. [57] found a reduction in school activities (including homework), over 50% of the sample in the study by Szpunar et al. [78] reported

that time spent on homework increased during the pandemic. Both studies had a high sample size at a national level (over 3000 participants in the first, and more than 5000 in the second), although both with a small response from the southern regions of Italy. This disparity in results could be due to a difference in the definition of the concept of "school activities", presented within the respective questionnaires or interpreted by participants differently, leading to different results. Both studies reported a dramatic decrease in extracurricular activities. Only two studies assessed the emotional aspects of students according to teachers [51, 65] and a single study by the students themselves [77]. However, their findings did not differ from those of most studies with parents in their sample. The students evaluated by Segre et al. [77] reported having difficulty getting used to distance learning, experiencing conditions of psychological distress and increased levels of anxiety. Both Crescenza et al. [65] and Doz et al. [51] highlighted the impoverishment of relationships, the difficulty of keeping the motivation of their students high and the difficulty in offering an effective learning environment to them. In general, our systematic review reveals a general agreement on the fact that the new way of learning had drastically changed the daily habits of students, with repercussions on their emotional states and ability to pay attention during school and extracurricular activities. All stakeholders expressed concern about the long-term effects of distance learning, including attention and communication difficulties, decreased motivation, mood changes (in particular, aggressiveness and restlessness), and learning difficulties.

**Marginality-inequality effects.** Several studies [44, 49, 53, 79] focused on children with neurodevelopmental disorders, finding that they experienced declines in various learning areas. It turns out that these students have struggled with distance learning due to the need for personalized education plans, which resulted to be difficult to implement online [76]. The feeling of difficulty for distance learning was also shared by parents and teachers. Parents generally considered the school support provided to their children to be insufficient; while teachers had difficulty engaging students with special educational needs [62, 66, 76]. In some cases, these students did not even participate in the lesson itself, due to complications in carrying out their distance education plans [58, 64, 71] or the absence of adequate technological tools on the part of the family unit [75], further worsening their learning [44, 58] or health condition [79]. The complications experienced in a new home learning environment were similar in an environment such as the School-in-Hospital Service [59], where teachers of hospitalized children encountered both relational and organizational difficulties teaching. However, two studies [72, 73] did not find any worsening condition during distance learning among students at risk of marginalization compared to the pre-pandemic period. Inguscio et al. [72] evaluated the academic well-being during distance learning of students between 8 and 19 years old with and without hearing loss. Marchese et al. [73] studied the effectiveness of distance learning in a sample of primary (and secondary) school students with specific learning disabilities and special educational needs. This difference with most of the studies identified could be due to the sample size of both studies, which is not representative of the population under consideration. Inguscio et al. [72] presented a sample of 41 students with hearing loss out of a total of 61, of which 21 were primary school students (14 with hearing loss). The study by Marchese et al. [73] involved 21 students from a single primary school, in southern Italy. We found further heterogeneous results even about the progressive worsening of the academic skills of students with socio-economic difficulties. Borgonovi & Ferrara [47], in line with Contini et al. [48] and Termine et al. [79] highlighted that the pandemic has exacerbated existing educational inequalities, particularly for students from lower socioeconomic backgrounds, with some Italian provinces and regions more affected than others. However, Bazoli et al. [46] classified students based on various social factors and found that students from disadvantaged backgrounds did not necessarily experience the most significant decline in learning outcomes. The diversity of these results may be due to the authors' use of different methodology utilized for their investigations. In

general, the data reflects a more widely shared opinion that the experience of distance learning has increased the difficulties of students who were already experiencing a difficult situation, worsening educational inequalities.

**Didactic problems.** Considering the challenges highlighted by various studies included in our systematic review, we can draw the conclusion that an efficient local network infrastructure can play a significant role in mitigating the negative effects of potential school closures, although not the only one. Our findings, in fact, revealed that many parents and teachers faced challenges related to digital skills and digital literacy [51, 52, 58, 68, 70, 76]. Furthermore, a common problem across all studies was that a large percentage of teachers reported they did not receive adequate training in the new teaching methodologies linked to distance learning or in the use of digital devices for teaching. Despite the challenges, some teachers saw positive aspects of digital learning, such as the potential for innovation, making education more open to technologies and student-centered learning processes [61, 66, 67, 75].

**Learning quality.** There was a general learning loss, particularly in mathematical skills, with greater damages for younger students (the studies did not consider primary school only), females and students with learning difficulties. All studies that considered a parental sample [57, 69, 76] agreed with the opinion of most teachers [61, 66] according to which distance learning did not improve students' academic performance. This is confirmed by studies that have taken into consideration the INVALSI performance of primary schools [46–48]. These studies highlighted a slight improvement in reading but a significant loss in learning mathematics compared to peers in the pre-pandemic period. Two studies show additional relevant considerations: Capperucci et al. [61] compared the academic performance of primary school students (according to the teachers' opinion) between two Italian regions (Tuscany and Umbria) and the national sample. From their analyses, it appeared that teachers who use "alternative" assessment methods had a higher evaluation in terms of students' academic performance. Again, Bertoletti et al. [52] highlighted how student performance varied based on how teachers themselves welcomed new learning possibilities and exploited their digital skills. The authors consider this information fundamental, underlining once again the importance of the learning environment and its emotional and cognitive effect on students [17].

As shown in Fig 2, most of the included studies (N = 22) dealt with the effects of school closures on the most fragile students, including students with specific diagnoses and/or sociocultural difficulties. In contrast, only nine studies focused on the academic performance of Italian students. This numerical disparity could be due to the pandemic condition, that has visibly exacerbated the condition of students already on the margins of education, and, among many possible reasons, the study methods created specifically for them could be not only ineffective but not even feasible in online mode. The low number of papers on academic performance, on the other hand, may be due to the difficulty for researchers to investigate electronically (online survey) directly on a primary school sample, which necessarily requires signed parental consent. In the same way, it is difficult for young students themselves to evaluate their personal academic performance, which is more of interest (particularly at this age) to their parents and teachers. In fact, a large part of the studies included in the learning quality section (specifically, those that investigated the construct directly on a sample of primary school students) used the data provided by INVALSI.

### Research Questions #2: Is there agreement between the opinions of parents, teachers, and students regarding the different effects of school closures on Italian primary school students?

Our results highlight several similarities and differences among the perspectives of parents, teachers, and students regarding the effects of COVID-19 school closures on students' learning

and well-being. A key similarity concerns the negative impact on learning and emotional well-being during the transition to remote learning. All three groups acknowledged a decline in students' attention, motivation, and engagement during online classes, as well as an increase in anxiety levels and concerns. Additionally, both parents and teachers highlighted the significant challenge in providing necessary support to students during remote learning, especially for students with special educational needs or neurodevelopmental disorders. This suggests an agreement among the perceptions of different stakeholders regarding the difficulties encountered by students during the pandemic. Such disadvantageous condition is not identified in students with hearing loss [72], neither according to the opinion of the students themselves nor their parents.

However, there are also differences in perspectives related to the main problems due to COVID-19. For example, parents expressed concerns about excessive screen time and decreased participation in extracurricular activities, while teachers mainly focused on the need to improve digital competence and the challenge of adapting assessment methods. Moreover, teachers acknowledged some advantages of remote learning, such as innovation in teaching methods, which parents seem not have directly noticed.

As previously mentioned, no study investigated potential difficulties among students in approaching the new instructional methodology. One possible explanation could be attributed to their adaptability to technology. It is plausible that these students, who are typically exposed to various digital devices and platforms from an early age, may have a propensity to navigate and engage with online learning tools. The early exposure to technology may have facilitated a smoother transition to distance learning, potentially reducing the perceived challenges or obstacles encountered during the adaptation process [86]. However, further research is needed about this point.

Students also highlighted specific difficulties related to their learning disorder and emphasized the negative impact on learning quality, thus supporting the conclusions of the other two groups (parents and teachers). It is interesting to note how both parents and teachers considered support for students crucial in this new mode of learning, although each stakeholders pointed to the other: parents were resentful of the teachers' lack of attention towards their pupils, especially those with previous difficulties; teachers were dissatisfied with the lack of attention and support from parents. Concerning the lack of support from some teachers, it is conceivable as expressed by some studies considered [52, 66]: teachers with more years of professional experience, accustomed to certain learning methodologies, encountered greater difficulties in reinventing themselves, especially towards technological novelty. The reported lack of parental support by teachers could be related with the level of education highlighted by Contini et al. [48], which suggest that students who have been most affected by COVID-19 school closures are those with highly educated parents. It is possible to postulate retrospectively that, the higher the level of education attained, the greater the professional responsibilities achieved by parents, and therefore the greater the workload, reducing the daily available time and hindering careful instructional support for their children. However, it is important to note that the relationship between parents' level of education and their involvement in their children's learning is complex and can vary depending on a range of factors, including cultural, socioeconomic, and individual context [30, 87].

### Research Questions #3: Are there any differences between the effects of school closures on Italian primary school students and students in other countries as appears in the international literature on the topic?

Finally, we examined eventual differences between the effects of school closures on Italian primary school students and the international literature on the topic. This inquiry was thoroughly

conducted through the analysis of the opinions of different populations (parents, teachers, students), while maintaining a focus on students' learning and well-being. This comparison could provide valuable insights into the specificities of the Italian experience and its relevance in the global context, thus contributing to delineating targeted strategies and interventions to address the effects of the pandemic on students' education.

Concerning the emotional aspects, our findings are in line with research conducted overseas. In their international systematic review, Viner et al. [24] analyzed the association between school closure and the well-being of K19 students. Compared to primary school students, school closure and the obligation to stay at home drastically reduced healthy extracurricular activities, such as sports, increasing sedentary habits (more time spent in front of the screen). Furthermore, a relationship between school closure and an increase in emotional and behavioral difficulties is once again confirmed. Our results are also in line with a recent systematic review of Italian K12 students during school closure [88]. Their results confirmed an increase in emotional dysfunction demonstrated by an increase in anxiety, anger, difficulty concentrating, and even sleep disturbances. However, it is interesting that Picca et al. [74] was the only study that found a positive effect of the pandemic period, precisely on children's sleep. Further research on this matter is necessary. Still, Panchal et al. [89] carried out an intercontinental systematic review on the mental health of children and adolescents ($\leq$ 19 years) exposed to the pandemic lockdown. Like them, our national data indicate that lack of adequate extracurricular activities and increased time spent in front of electronic tools negatively impacted children's emotional health and psychological well-being, increasing perceptions of negative emotions associated with school closures for COVID-19 (feeling of loneliness, anger, irritability, fear, and stress). Furthermore, the authors also identified a higher risk of mental health problems and increased stress for students with special educational needs, although this also depended on space availability (e.g., having a garden or adequate space for studying at home) and family socio economic status. Further international research highlighted how students already on the margins of school community suffered scholastically [13, 14] and emotionally because of the school closure more than their peers [15, 16]. Even in this area, our results do not differ from international literature, although we highlighted that the condition differed based on the difficulty experienced by the students (e.g., students with special education needs vs. low socio-economic status vs. linguistic-cultural differences).

We did not find international reviews that specifically addressed technical-practical educational issues, such as the knowledge and competence of teachers in adapting to new teaching methodologies or the availability and quality of technological devices in the context of distance learning. However, a study conducted by Engzell et al. [30] sheds some light on this matter. This study reported that Dutch schools were closed for "only" eight weeks during a period of pandemic-related disruptions, without experiencing complications in the continuation of education. One of the factors that contributed to the maintenance of an adequate education was the excellent quality of broadband in the country. As regards digital preparation of teachers in this new teaching method, our results are aligned with a survey conducted in Germany by König et al. [90], about the importance of teachers possessing digital competence in achieving meaningful educational objectives. In summary, while technical-practical issues were not extensively explored in secondary studies, the primary evidence, and data available do underscore the significance of factors such as broadband quality and teachers' digital skills in addressing the challenges of distance learning effectively.

As regards academic achievement, our review showed a learning loss in mathematical skills, in particular for females and students with learning difficulties, and a slight improvement in reading skills [8, 10–12]. A comparison with international review is needed due to the several differences and similarities between the Italian context and other countries. Contrary to Italian context, the international study by Betthäuser et al. [11] did not find an effects across age (no

difference between primary and older students in learning loss). Similarly to our results, they found a greater learning loss in math than reading, although we highlighted a slight improvement in reading skills of Italian primary students. Our data are in line with the Danish study by Birkelund & Karlson [91], that found a leaning gain in reading performance of pupils in Grades 2 and 4, and a leaning loss in the same topic between older students (grade 8). It is important to note they lived a different time range of school closure: only eight weeks for Danish primary school students, and 22 weeks for Danish Grade 8 students. Hence, it is conceivable that prolonged school closures may have a detrimental effect on children's learning (see for example Ref. [92]), raising long-term concerns for countries that experienced prolonged school closures, such as Italy. Our data are also in line with the US study by Kuhfeld, Lewis & Peltier [93], where they highlighted a larger decline for primary school students compared to secondary school students, but not in the same school subjects as the ones that were analyzed in the current review: while Italian studies highlighted greater learning loss in mathematics and a slight improvement in reading, they found similar pre- and post-pandemic results for mathematics, and a greater loss in reading among younger students. Furthermore, our study highlighted a worsening in existing disparities among students, especially those with learning disabilities or special education needs (SEN), and from lower socioeconomic backgrounds. These results are aligned with several international studies. As regards students with previous scholastic difficulties, two US studies [94, 95] highlighted that learning loss in reading and mathematics was substantially greater for students with prior difficulties than for the general population. As previously reported, it is conceivable that the structured education provided by the schools that these students require could not be adequately delivered in the new learning modality. This once again underlines the need to identify new, more inclusive learning methodologies. Compared to students with different family backgrounds, heterogeneity is highlighted at an international level. While the study by Kuhfeld, Lewis & Peltier [93], in line with our findings, highlighted considerable differences between ethnic groups, with lower performance among students with high socioeconomic poverty, not all studies found an association between learning loss and socioeconomic status [91]. We might assume that this difference is due to the concept of "socio-economic status" varies considerably from country to country (in this case: USA vs. Denmark). This may mean that measures used to assess the socioeconomic level of study participants may not be perfectly comparable across different nations or regions. Furthermore, country-specific cultural, political, and economic factors could significantly influence how socioeconomic level affects student learning losses. Furthermore, it should be remembered that the sample of the Danish primary school of Birkelund & Karlson [91] experienced a school closure of only 8 weeks, while the study of Kuhfeld, Lewis & Peltier [93] referred to US students, where the school closure lasted several months. This would further confirm the hypothesis that the rapid return to school reduced the risk of learning losses for students with previous difficulties, also in relation to the possible lack of adequate devices to follow distance learning (e.g., electronic devices; good quality or total absence of Wi-fi) within the family unit. The comparison between Italian and international studies highlighted an extremely heterogeneous picture, with numerous differences and similarities that showed a more or less harmful effect of school closures due to COVID-19 based on different factors: territory; duration of school closure; pre-pandemic disadvantaged conditions. Further research on the topic is needed to highlight long-term post-pandemic effects on the quality of learning of primary school students, not only Italians.

## Limitations

The authors are aware of the limitations of this systematic review. First, although a good number of studies have been identified, few of them have achieved a satisfactory level of risk of

bias. This may be attributable to the fact that, due to the pandemic condition, most of the studies took place through the completion of online questionnaires and snowball sampling, and in some cases validated measures were not used to evaluate the pandemic effects of interest to them. Also, except for Baschenis et al. [44], all studies were observational, and most were cross-sectional. A further limitation is, paradoxically, the number of studies. Taking into consideration all the included studies, an adequate number of 34 papers is reached. However, having examined 4 distinct and fundamental categories of learning during the pandemic period and as many as 3 different samples (parents, teachers, pupils), the number of studies (divided according to section and reference sample) has never exceeded the number of 9 papers. Hence, further research is needed regarding the perception of distance learning and its consequences on learning by all stakeholders. Moreover, the substantial differences in the analysis of the results (sometimes even absent) of the studies identified did not allow the development of a meta-analytic study. Consequently, the qualitative nature of the studies included in this systematic review, did not permit the evaluation of a publication bias, aimed exclusively at evaluating a bias within quantitative studies (therefore, meta-analyses). However, an effort was done so that to avoid publication bias, following a rigorous research methodology, including PRISMA flowchart and checklist, a Risk-of-Bias Quality Assessment, and a search strategy. Another important limitation of our study is that it was not always possible to quantify the differences between the reference sample and the pre-pandemic period. Most of the studies, in fact, are based on the opinion of students, parents and/or teachers about the differences between the two time periods, collecting these values through surveys and reporting values in percentages. However, it has not always been possible to draw a comparison between the two periods. With respect to academic performance, the studies in this regard mainly focused on INVALSI data, relating to reading, mathematics, and English, although no identified study reported information on the latter school subject. Consequently, it was not possible to identify a change–positive and/or negative–with respect to all primary school subjects.

## Conclusions

Although further research is needed, our systematic review would like to offer an overview of the effects of the school closure due to COVID-19 on primary school students, with reference to the Italian territory, among the most affected by the pandemic. Our results show that more attention needs to be paid to the emotional needs of students, especially after the distance learning period, focusing on children at risk. Distance learning has also had an impact on parents/teachers' remote working, leading to an increase in difficulties within the family and, above all, impacting the learning context. The lack of adequate interventions from the early years of education could increase the risk of amplifying territorial inequalities from the outset, as highlighted by the recent INVALSI data [96]. There is a clear decline in the performance of primary school students (assessed in the second and fifth school years) in all subjects examined (Italian, Mathematics, English). This indicates that the pandemic's effect on learning has not yet diminished. Furthermore, there is evidence of territorial disparities, with southern Italian regions showing a significantly more pronounced gap and lower learning conditions for students with a migrant background. It therefore becomes essential to implement new learning methodologies, not only to monitor children's learning outcome but also to involve them in the learning process and ensure a real acquisition of knowledge, even remotely. It is therefore necessary to implement changes at a national level starting, at a technical-practical level, from the improvement of the quality of the Italian broadband and from the economic support to families who cannot afford adequate digital tools for remote study of their child.

It is also of significant importance to counteract the decline in students' academic motivation, by paying greater attention to students' emotional needs, using practices that aim at the expression of their needs and experiences during such a period full of emotional conflicts. These practices must, in turn, be directed towards families and teachers, who themselves experienced a challenging and non-peaceful condition during the school closure due to the pandemic. The importance of collaboration between educational institutions and families is underlined, to create a network of emotional and academic support. In this sense, the increasingly stable implementation of the figure of the school psychologist in schools is considered relevant, for a synergistic work of the entire educational community. Furthermore, the need and relevance of continuous training programs for teachers, aimed at to the promotion of useful skills for the challenges related to distance learning. Last, it is necessary to reformulate the Individualized Educational Plans and Personalized Didactic Plans of students with specific needs: the current face-to-face nature of these programs has not allowed a real involvement of students in the class group, preventing inclusion. We are aware that this is a great innovative challenge, nevertheless necessary for a real recovery of learning in not only quantitative but also qualitative terms. Our work further highlights the need for coordinated efforts among educational institutions, local and national governments, as well as society as a whole to promote a more effective, equitable, and inclusive educational system.

## Supporting information

**S1 Checklist. PRISMA 2020 checklist.**
(PDF)

**S1 Fig. ROBINS-I tool.**
(TIFF)

**S1 Table. Newcastle-Ottawa scale adapted for cohort studies (NOS-C).**
(PDF)

**S2 Table. Newcastle-Ottawa scale adapted for case-control studies (NOS-CC).**
(PDF)

**S3 Table. Newcastle-Ottawa scale adapted for cohort studies (NOS-C).**
(PDF)

## Author Contributions

**Conceptualization:** Eugenio Trotta, Gianluigi Serio, Lucia Monacis, Chiara Valeria Marinelli, Annamaria Petito, Tiziana Quarto, Paola Palladino.

**Data curation:** Eugenio Trotta, Gianluigi Serio, Aurora Bonvino, Antonella Calvio, Roberta Stallone, Tiziana Quarto, Paola Palladino.

**Formal analysis:** Eugenio Trotta, Gianluigi Serio, Aurora Bonvino, Antonella Calvio, Roberta Stallone, Tiziana Quarto, Paola Palladino.

**Methodology:** Eugenio Trotta, Gianluigi Serio, Lucia Monacis, Leonardo Carlucci, Chiara Valeria Marinelli, Annamaria Petito, Giovanna Celia, Aurora Bonvino, Antonella Calvio, Roberta Stallone, Ciro Esposito, Stefania Fantinelli, Francesco Sulla, Raffaele Di Fuccio, Gianpaolo Salvatore, Tiziana Quarto, Paola Palladino.

**Supervision:** Lucia Monacis, Leonardo Carlucci, Chiara Valeria Marinelli, Annamaria Petito, Giovanna Celia, Aurora Bonvino, Antonella Calvio, Roberta Stallone, Ciro Esposito,

Stefania Fantinelli, Francesco Sulla, Raffaele Di Fuccio, Gianpaolo Salvatore, Tiziana Quarto, Paola Palladino.

**Writing – original draft:** Eugenio Trotta, Gianluigi Serio, Francesco Sulla, Tiziana Quarto, Paola Palladino.

**Writing – review & editing:** Eugenio Trotta, Gianluigi Serio, Lucia Monacis, Leonardo Carlucci, Chiara Valeria Marinelli, Annamaria Petito, Giovanna Celia, Ciro Esposito, Stefania Fantinelli, Francesco Sulla, Raffaele Di Fuccio, Gianpaolo Salvatore, Tiziana Quarto, Paola Palladino.

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
