## [Decision Letter · Decision Letter 0]

24 Jan 2024

PONE-D-23-43290The effects of the COVID-19 pandemic on Italian primary school children’s learning: A systematic review through a psycho-social lensPLOS ONE

Dear Dr. Palladino,

Thank you for submitting your manuscript to PLOS ONE. After careful consideration, we feel that it has merit but does not fully meet PLOS ONE’s publication criteria as it currently stands. Therefore, we invite you to submit a revised version of the manuscript that addresses the points raised during the review process.

**ACADEMIC EDITOR: **Dear authors, despite the technical and formal quality of the review performed, as you can see from the reviewers' comments, the paper is not informative enough and still does not contribute much to the current literature and knowledge. However, if you are willing to change the way you analyse the results, taking into account the reviewers’ suggestions, we could re-evaluate the paper. Please also bear in mind how much effort and time you will need to do this, as the changes requested are quite extensive.

We look forward to receiving your revised manuscript.

Kind regards,

Laura Brunelli, MD, PhD

Academic Editor

PLOS ONE

Journal Requirements:

2. Please ensure that you include a title page within your main document. You should list all authors and all affiliations as per our author instructions and clearly indicate the corresponding author.

3. Please include your tables as part of your main manuscript and remove the individual files. Please note that supplementary tables (should remain/ be uploaded) as separate ""supporting information"" files.

Reviewers' comments:

Reviewer's Responses to Questions

**Comments to the Author**

1. Is the manuscript technically sound, and do the data support the conclusions?

Reviewer #1: Partly

Reviewer #2: Partly

2. Has the statistical analysis been performed appropriately and rigorously? 

Reviewer #1: N/A

Reviewer #2: N/A

3. Have the authors made all data underlying the findings in their manuscript fully available?

Reviewer #1: Yes

Reviewer #2: Yes

4. Is the manuscript presented in an intelligible fashion and written in standard English?

Reviewer #1: Yes

Reviewer #2: Yes

5. Review Comments to the Author

Reviewer #1: The article is well written and scientifically sound. The topic is very relevant to decision makers and pubblic health professionals, however the aims appear too broad and the study population too heterogeneous. I understand the intention of the authors to be as inclusive in the study selection as possible given the circumstances and the limitations of the source litterature, but examinig at the same time three different populations (Students, teachers and parents) makes difficult for the reader to appraise whom the raccomandations are directed. I suggest to change the study design from systematic review to scoping review, see this article by Munn et al. for guidance on the difference between scoping and systematic reviews (https://doi.org/10.1186/s12874-018-0611-x). The results are well presented and synthesize effectively the studies analyzed, however this review lacks of an analysis of the pubblication bias of the studies included. I suggest the authors to add a section about pubblication bias and its impact on the results of this review.

Reviewer #2: The authors present a systematic review on how Covid-related school closures affected Italian primary education. The literature search is well-documented and seems to have been competently executed. A risk-of-bias analysis has been done according to the standards of the field. Nevertheless, I cannot recommend publication: we simply do not learn very much from the review.

This is mostly a result of the way the included studies are analyzed. Or not analyzed, because studies are only listed and summarized. There is little attempt at synthesis, comparison and contrast. Thus, the review mostly adds to the literature by being a bibliography., not as an original contribution.

It is also not easy for the authors:

-the authors mostly found surveys that elicited opinions from parents and teachers, and these hardly lend themselves to meta-analysis.

-there are already ample reviews on the pandemic's effects on education. We know that it was hard on all parties, that transitioning to distance learning was difficult, and that school closures resulted in learning losses and emotional problems. The authors summarize this competently in the introduction, and then summarize some 40 studies in which parents, teachers (and, in some studies, students) lament that it was hard on all parties, transitioning to distance learning was difficult, and that school closures resulted in learning losses and emotional problems. The only thing we learn from this is that Italian parents, teachers and students were not in fact deluded about the pandemic's effect on education. Which we already learned from the studies themselves.

So how to make the study contribute? If the authors could find some way of integrating and contrasting the findings from the studies, this might lead to an interesting paper. E.g., do the three parties (parents, teachers, students) agree on everything? On what do they disagree? Do they have diagnoses that might not be known in the literature?

As an additional point, some proofreading is in order. Some sentences are odd. Examples from the first page:

-"dramatically changed people’s life worldwide, impacting all sectors, with evidence for the school sector". What does this "with evidence" mean?

-"DL is a process that needs a careful goal design". Typically, distance learning (better not abbreviate such terms, makes text harder to read) needs goals and needs a careful design. A goal design would seem to refer to a football field.

-"extemporaneousness that". extemporaneous is already a rare, difficult word but the noun simply does not exist

-Bayley et al...: there is quite an extensive discussion of forecasting of Covid effects while the pandemic raged, but that is not needed anymore now that we know the outcomes - unless the idea is that we should congratulate the forecasters.

6. PLOS authors have the option to publish the peer review history of their article (what does this mean?). If published, this will include your full peer review and any attached files.

Reviewer #1: **Yes: **Marco Driutti

Reviewer #2: No

---

## [Author Response · Author response to Decision Letter 0]

22 Mar 2024

We thank the Academic Editor and Reviewers for the thorough revisions of our work and for the constructive suggestions provided. Responses to comments in this letter are marked in bold. As requested, we included a marked-up copy of our manuscript that highlights changes made to the original version, called 'Revised Manuscript with Track Changes'.

ACADEMIC EDITOR:

Dear authors, despite the technical and formal quality of the review performed, as you can see from the reviewers' comments, the paper is not informative enough and still does not contribute much to the current literature and knowledge. However, if you are willing to change the way you analyze the results, taking into account the reviewers’ suggestions, we could re-evaluate the paper. Please also bear in mind how much effort and time you will need to do this, as the changes requested are quite extensive.

Authors’ response: 

Dear Professor Brunelli, we sincerely appreciate the thorough and constructive review conducted by the Reviewers. Their comments provided valuable feedback on the technical and formal aspects of our paper. In response to your and their suggestions, we committed to revisiting our approach to result analysis and to incorporating the recommended changes. We value this opportunity to address the reviewers' concerns, and we thank you for your consideration in re-evaluating our submission. We assure you that we made every effort to implement the suggested modifications in a timely and effective manner. 

Review Comments to the Author

Reviewer #1

Comments to the Author #1: The article is well written and scientifically sound. The topic is very relevant to decision makers and public health professionals, however the aims appear too broad and the study population too heterogeneous. I understand the intention of the authors to be as inclusive in the study selection as possible given the circumstances and the limitations of the source literature, but examining at the same time three different populations (Students, teachers and parents) makes difficult for the reader to appraise whom the recommendations are directed.

Authors’ response #1: 

Dear Professor Driutti, many thanks for your appreciation and positive evaluation of our work. About the heterogeneity of the studied population, we would like to specify that in our work the studied population as well as the target population of our results and recommendations is the Italian primary school students’ population. Teachers and Parents are only considered for their role as reference figures for the students, thus included as vehicles of opinions and information about the effects of school closure due to COVID-19 on students. We understand that this point was not clear in the previous version of the manuscript, thus we have now better specified it, particularly in the Aim section in which we have now added a clear reference to the students’ population as the target population. 

Moreover, we also agree with Dr. Driutti that the previous version of the manuscript did not well clarify the aims of the study, which appeared “too broad” to the reader. Therefore we made a series of changes in the manuscript according to this comment. First, we modified the section called ‘Aim of the present study’ (see page 6), reporting the three research questions we intended to answer, namely:

1. Research Questions #1: Beyond academic performance, are there other dimensions of Italian primary school students affected by school closures?

2. Research Questions #2: Is there agreement between the opinions of parents, teachers and students regarding the different effects of school closures on Italian primary school students?

3. Research Questions #3: Are there any differences between the effects of school closures on Italian primary school students and students in other countries as appears in the international literature on the topic?

Starting from an overview of the effects of the school closure due to the COVID-19 pandemic, we explained the identified dimensions affected by it, providing a description of them (see page 1 of the manuscript).

We reformulated the Results starting from the population: instead of dividing first by dimension and then by population, we reported the results found for the specific population (always with a focus on Italian primary school students), including their opinion on all dimensions (e.g.: Parents' opinion - see pages 16-27).

We answered research questions in our Discussion (see pages 28-36).

We believe that these changes in paper structure also contribute to answer to the reviewer’s concern about the studied populations and aim’s comprehension.

Comments to the Author #2: I suggest to change the study design from systematic review to scoping review, see this article by Munn et al. for guidance on the difference between scoping and systematic reviews (https://doi.org/10.1186/s12874-018-0611-x). 

Authors’ response #2: 

We studied carefully the interesting article that you shared with us. After several and careful considerations, we decided to maintain our review design as a systematic review according to the following points As reported By Munn et al. (2018), citing Higgins, Altman & Sterne (2011), a systematic review ‘uses explicit, systematic methods that are selected with a view to minimizing bias, thus providing more reliable findings from which conclusions can be drawn and decisions made’. We believe that the previous version of the manuscript respected the research method typical of systematic reviews, but did not satisfy the decision-making through the answer to certain research questions, as you and the Reviewer #2 correctly pointed out. Therefore, we modified the section called ‘Aim of the present study’, reporting the three aforementioned research questions. Furthermore, we modified the structure of our Results and Discussion to highlight the findings regarding each research question. Thus, we believe that the new version of the manuscript fully respects all criteria for being defined as systematic review.

Comments to the Author #3: The results are well presented and synthesize effectively the studies analyzed, however this review lacks an analysis of the publication bias of the studies included. I suggest the authors add a section about publication bias and its impact on the results of this review.

Authors’ response #3: 

We thank the Reviewer for raising this point. We acknowledge the relevance of a funnel plot that graphically describes an hypothetical publication bias. Unfortunately, the qualitative nature of the studies included in this systematic review, did not afford the evaluation of a publication bias, aimed exclusively at evaluating a bias within quantitative studies (therefore, meta-analyses). We are not aware of, and have not been able to find in the scientific literature, a method of evaluating publication bias in the case of qualitative studies. However, we understand the possibility of any bias, so we reported a series of Risk-of-Bias assessments, based on study design (See “Methods” for a description of the process, “Results” for evaluations, and “Supporting information” file for the tables of Risk-of-Bias). Furthermore, we tried to avoid publication bias, following a rigorous research methodology, including a PRISMA flowchart and checklist, the mentioned Quality Assessment (Risk-of-Bias), and a search strategy. We thank again the Reviewer for raising this point, giving us the opportunity to better specify the nature of the studies identified (see page 38).

Reviewer #2

Comments to the Author #1: The authors present a systematic review on how Covid-related school closures affected Italian primary education. The literature search is well-documented and seems to have been competently executed. A risk-of-bias analysis has been done according to the standards of the field. Nevertheless, I cannot recommend publication: we simply do not learn very much from the review. This is mostly a result of the way the included studies are analyzed. Or not analyzed, because studies are only listed and summarized. There is little attempt at synthesis, comparison and contrast. Thus, the review mostly adds to the literature by being a bibliography., not as an original contribution.

Authors’ response #1:

We sincerely appreciate the Reviewer’s comments on our manuscript. His/Her feedback is invaluable to us, and we have carefully considered his/her comments. We acknowledge his/her concerns regarding the analysis of included studies and the need for more synthesis, comparison, and contrast. We took these observations seriously and endeavored to improve our work accordingly. In our revised manuscript, we made substantial efforts to enhance the analysis of the included studies, providing a more comprehensive synthesis and addressing the limitations highlighted.

We understand the importance of offering a meaningful contribution to the literature, and we are committed to ensuring that our work not only serves as a bibliography but also provides valuable insights. We genuinely believe that the revisions made have strengthened the overall quality of the paper. We kindly request the reconsideration of our manuscript, taking into account the modifications made in response to his/her thoughtful feedback. We are hopeful that the revised version addresses his/her concerns and merits a more positive evaluation.

We thank the Reviewer #2 once again for his/her time and constructive criticism. We greatly appreciate his/her dedication to maintaining the high standards of the field.

 Comments to the Author #2: It is also not easy for the authors:

-the authors mostly found surveys that elicited opinions from parents and teachers, and these hardly lend themselves to meta-analysis.

-there are already ample reviews on the pandemic's effects on education. We know that it was hard on all parties, that transitioning to distance learning was difficult, and that school closures resulted in learning losses and emotional problems. The authors summarize this competently in the introduction, and then summarize some 40 studies in which parents, teachers (and, in some studies, students) lament that it was hard on all parties, transitioning to distance learning was difficult, and that school closures resulted in learning losses and emotional problems. The only thing we learn from this is that Italian parents, teachers and students were not in fact deluded about the pandemic's effect on education. Which we already learned from the studies themselves.

So how to make the study contribute? If the authors could find some way of integrating and contrasting the findings from the studies, this might lead to an interesting paper. E.g., do the three parties (parents, teachers, students) agree on everything? On what do they disagree? Do they have diagnoses that might not be known in the literature?

Authors’ response #2:

We appreciate the comment, admitting that it was not easy for us. As He/She reported, we found mostly qualitative studies with surveys, and a snowball recruitment. This, together with the lack of statistical data, led us to opt for a systematic review instead of a meta-analysis, thus adapting our investigation to the nature of the included studies, but preserving the rigor of the methodology we intended to use to investigate and report results. 

We are aware that there are already numerous reviews on the effects of the pandemic on education. However, we believe that our research has many points of originality and it has the potential to add new and important information to the literature in the field. First, our systematic review is the only review addressing the pandemic’s effect on education on an Italian primary students population. This represents an important adding point to the literature not only because of the difference in nationality per se, but also because the investigated phenomenon impacted Italy, with respect to all the other European countries, in a peculiar and severe manner that made Italy a unique source of investigation of the impact of this phenomenon on education, regardless of the nationality. Thus, we strongly believe that the Italian point of view in this field is not just one other point of view together with the one of other countries, but it has a dramatic privilege to be the point of view of the European country in which children experienced the longest school closures and one of the greatest Covid-19 impact in terms of deaths and infections in the first pandemic months. Telling about the impact of Covid-19 pandemic on the Italian students' population means telling the international literature about the European country in which this tragic event has potentially had the greatest effect. Moreover, our review is the only review in the field that takes into account such an extensive set of learning-related dimensions, rather than focusing only on the academic achievement. Also, by including studies that explore the opinion of teachers, parents and students themselves, this systematic review constitutes the opportunity to observe the same phenomenon by different perspectives which is a rarity in the field.

But we absolutely agree with the reviewer that these original points and new information were not correctly marked in the previous version of the manuscript. Thus, we have now improved the quality of our work, marking the points of originality throughout the manuscript. Importantly, following the specific reviewer’s indication, we re-analysed the results of the included studies in order to answer specific research questions, also adding elements of originality in addition to the previously reported ones. In particular, we modified the section called ‘Aim of the present study’ (see page 6), reporting the three research questions we intended to answer, namely:

1. Beyond academic performance, are there other dimensions of Italian primary school students affected by school closures?

2. Is there agreement between the opinions of parents, teachers and students regarding the different effects of school closures on Italian primary school students?

3. Are there any differences between the effects of school closures on Italian primary school students and students in other countries as appears in the international literature on the topic?

Furthermore, we modified the structure of our Results and Discussion to highlight the findings regarding our aforementioned research questions (see pages 28-36).

 Comments to the Author #3: As an additional point, some proofreading is in order. Some sentences are odd. Examples from the first page:

-"dramatically changed people’s life worldwide, impacting all sectors, with evidence for the school sector". What does this "with evidence" mean?

-"DL is a process that needs a careful goal design". Typically, distance learning (better not abbreviate such terms, makes text harder to read) needs goals and needs a careful design. A goal design would seem to refer to a football field.

-"extemporaneousness that". extemporaneous is already a rare, difficult word but the noun simply does not exist

-Bayley et al...: there is quite an extensive discussion of forecasting of Covid effects while the pandemic raged, but that is not needed anymore now that we know the outcomes - unless the idea is that we should congratulate the forecasters.

Authors’ response #3:

We thank the Reviewer for his/her suggestions, and we apologize for the grammatical errors and the word usage. Before the submission, the whole manuscript was proofread by a native English colleague, expert of the field. However, after the Reviewer’s error reporting, we agree that further and more careful proofreading is required. Therefore, the entire manuscript was once again brought to the attention of a native professional speaker. Regarding the examples of the Reviewer from the first page:

● “dramatically changed [...] with evidence for the school sector”. We changed “with evidence” to “with repercussions also for”;

● “DL is a process that needs a careful goal design”. Thank you for your suggestion. We changed all DL acronyms to “Distance Learning” for the convenience of the reader. Furthermore, we changed “needs a careful goal design” to “needs a careful design”;

● thank you for your su

---

## [Decision Letter · Decision Letter 1]

6 May 2024

The effects of the COVID-19 pandemic on Italian primary school children’s learning: A systematic review through a psycho-social lens

PONE-D-23-43290R1

Dear Dr. Palladino,

We’re pleased to inform you that your manuscript has been judged scientifically suitable for publication and will be formally accepted for publication once it meets all outstanding technical requirements.

Kind regards,

Laura Brunelli, MD, PhD

Academic Editor

PLOS ONE

Additional Editor Comments (optional):

Reviewers' comments:

Reviewer's Responses to Questions

**Comments to the Author**

1. If the authors have adequately addressed your comments raised in a previous round of review and you feel that this manuscript is now acceptable for publication, you may indicate that here to bypass the “Comments to the Author” section, enter your conflict of interest statement in the “Confidential to Editor” section, and submit your "Accept" recommendation.

Reviewer #1: All comments have been addressed

Reviewer #2: All comments have been addressed

2. Is the manuscript technically sound, and do the data support the conclusions?

Reviewer #1: Yes

Reviewer #2: Yes

3. Has the statistical analysis been performed appropriately and rigorously? 

Reviewer #1: Yes

Reviewer #2: N/A

4. Have the authors made all data underlying the findings in their manuscript fully available?

Reviewer #1: Yes

Reviewer #2: Yes

5. Is the manuscript presented in an intelligible fashion and written in standard English?

Reviewer #1: Yes

Reviewer #2: Yes

6. Review Comments to the Author

Reviewer #1: (No Response)

Reviewer #2: My comments have been fully addressed. Most notably, there is now a new section where results from the different groups of participants are compared. It is a bit odd that this is part of the discussion instead of results, but this will not bother a reader interested in this topic.

7. PLOS authors have the option to publish the peer review history of their article (what does this mean?). If published, this will include your full peer review and any attached files.

Reviewer #1: **Yes: **marco driutti

Reviewer #2: No

---

## [Editor Report · Acceptance letter]

24 May 2024

PONE-D-23-43290R1 

PLOS ONE

Dear Dr. Palladino, 

I'm pleased to inform you that your manuscript has been deemed suitable for publication in PLOS ONE. Congratulations! Your manuscript is now being handed over to our production team.

Kind regards, 

on behalf of

Dr. Laura Brunelli 

Academic Editor

PLOS ONE